# Detecting image manipulation with ELA-CNN integration: a powerful framework for authenticity verification

Ahmad M. Nagm[1], Mona M. Moussa[2], Rasha Shoitan[2], Ahmed Ali[3,4], Mohamed Mashhour[1,5], Ahmed S. Salama[1,6] and Hamada I. AbdulWakel[5]

[1] Department of Computer Engineering and Electronics, Cairo Higher Institute for Engineering, Computer Science and Management, Cairo, Egypt
[2] Computer and Systems Department, Electronics Research Institute, Cairo, Egypt
[3] Department of Computer Science, College of Computer Engineering and Sciences, Prince Sattam Bin Abdulaziz University, Al-Kharj, Saudi Arabia
[4] Computer Science, Higher Future Institute for Specialized Technological Studies, Cairo, Egypt
[5] Computer Science Department, Faculty of Computers and Information, Minia University, Minia, Egypt
[6] Electrical Engineering Department, Faculty of Engineering & Technology, Future University in Egypt, New Cairo, Egypt



Corresponding author
Ahmed Ali,
a.abdalrahman@psau.edu.sa

## ABSTRACT

The exponential progress of image editing software has contributed to a rapid rise in the production of fake images. Consequently, various techniques and approaches have been developed to detect manipulated images. These methods aim to discern between genuine and altered images, effectively combating the proliferation of deceptive visual content. However, additional advancements are necessary to enhance their accuracy and precision. Therefore, this research proposes an image forgery algorithm that integrates error level analysis (ELA) and a convolutional neural network (CNN) to detect the manipulation. The system primarily focuses on detecting copy-move and splicing forgeries in images. The input image is fed to the ELA algorithm to identify regions within the image that have different compression levels. Afterward, the created ELA images are used as input to train the proposed CNN model. The CNN model is constructed from two consecutive convolution layers, followed by one max pooling layer and two dense layers. Two dropout layers are inserted between the layers to improve model generalization. The experiments are applied to the CASIA 2 dataset, and the simulation results show that the proposed algorithm demonstrates remarkable performance metrics, including a training accuracy of 99.05%, testing accuracy of 94.14%, precision of 94.1%, and recall of 94.07%. Notably, it outperforms state-of-the-art techniques in both accuracy and precision.

## INTRODUCTION

It has never been easy to create fake images or videos. In the contemporary era, the widespread availability of image processing tools has made image editing much easier for

individuals without extensive technical expertise. These range from conventional tools such as Photoshop to advanced neural network-based tools such as DeepFakes, which led to a rise in crimes such as identity theft, privacy invasion, and the spread of fake news. Copy-move and image splicing are considered the most common and significant techniques for image manipulation. In copy-move, a small portion of an image is copied and pasted into different areas within the same image. On the other hand, image splicing entails selecting a region from one image and pasting it into another where it appears suitable. Addressing copy-move and splicing forgeries is of the utmost significance in digital images, as these manipulations significantly impact numerous real-world settings. A manipulated image in the news media could influence public opinion regarding an event. Also, the manipulated image may be utilized to fabricate evidence for legal objectives. A fabricated image can potentially propagate false information or harm an individual's reputation, even in routine circumstances (*Jain & Goel, 2021*; *Machado et al., 2019*; *Maji et al., 2020*). Hence, detecting these types of manipulation is critical for establishing the credibility of digital images in the current day. It is nearly impossible for a person to always determine whether digital content is unaltered or altered. Thus, exploring more effective methods for identifying manipulated or fraudulent images is crucial (*Fu, Zhang & Wang, 2023*; *Hosny et al., 2023*; *Nagm & Elwan, 2021*).

The detection of copy-move forgery poses a significant challenge due to the inherent difficulty of distinguishing the forged region from the original image. This is primarily attributed to the fact that the statistical distribution of the image pixels remains largely similar, further complicating the identification process. Furthermore, splice detection methods involve examining several factors to identify instances of image forgery. These factors include the analysis of changes in the statistical distribution of pixels, variations in the compression level, and the examination of various features associated with the camera used to capture the image (*Ghannad & Passi, 2023*). Several researchers have suggested algorithms designed to identify copy-move and image splicing, progressing from traditional approaches to more advanced technologies like deep learning. Traditional methods often depend on manually designed features and rules to detect anomalies that suggest manipulation (*Alahmadi et al., 2013*; *He et al., 2012*; *Gani & Qadir, 2020*). However, modern approaches utilize deep learning techniques to automatically acquire and extract complex patterns and representations from extensive image datasets (*Meena & Tyagi, 2021*; *Tyagi & Yadav, 2023a*).

Until now, image manipulation research has remained open to developing models with high accuracy to detect copy-move and image splicing. Therefore, this research proposes a CNN-based approach for detecting image manipulations, specifically targeting splicing and copy-move forgeries. The proposed method combines error level analysis (ELA) with a CNN to effectively extract features that reflect the modified regions of the image. First, the ELA is employed to identify areas in the input image that have been tampered with, and then the ELA output is fed into the proposed CNN model to recognize the manipulations automatically. The proposed method utilizes a CNN model with specific components. It includes two convolutional layers, a max pooling layer, two dropout layers, and two fully connected layers. The contributions of this article are as follows:

- A novel image forgery detection algorithm that integrates ELA and CNN is proposed to detect copy-move and splicing forgeries in images.
- ELA is used as a preprocessing step to extract the tampering artifacts to improve CNN classification accuracy.
- A novel CNN model with two convolutional layers, one max-pooling layer, two dropout layers, and two fully connected layers is proposed for detecting image forgery.
- Extensive comparisons between the proposed model's results and existing models are made on the CASIA 2.0 dataset (*Dong, Wang & Tan, 2013*).

The subsequent sections of this research article are organized as follows: "Literature Review" presents an extensive review of prior studies. "Methodology" introduces the proposed framework architecture. "Experimental Results" outlines the dataset, evaluation metrics, experimental tests conducted, and a comparative analysis between the proposed model and state-of-the-art techniques. "Discussion" provides an ablation study for the model architecture, and "Conclusion" summarizes this research and its findings.

## LITERATURE REVIEW

Different hand-crafted and deep learning-based methods are proposed for detecting copy-move and image splicing. Hand-crafted methods utilize specialized forgery detection algorithms designed for copy-move detection. These algorithms employ various techniques, including block-based analysis (*Dua, Singh & Parthasarathy, 2020*; *Armas Vega et al., 2021*; *Meena & Tyagi, 2020*), keypoint analysis (*Badr, Youssif & Wafi, 2020*; *Alberry, Hegazy & Salama, 2018*), and noise or ELA (*Abd Warif et al., 2015*) to identify manipulated regions. While for image splicing, some of the most famous handcrafted features are discrete wavelet transform (DWT) (*He et al., 2012*), contourlet transform (CT) (*Zhang, Lu & Weng, 2016*), and discrete cosine transform (DCT) (*Alahmadi et al., 2013*; *Zhang et al., 2015*). *Khudhair et al. (2023)* present an approach for copy-move forgery detection that depends on partitioning an image equally into blocks. Singular value decomposition (SVD) is applied to each block, and a norm is selected to represent a scaling factor for the SVD of each block. The similar norms are then grouped, and according to the weight value, the image is classified as original or forgery. *Umamaheswari & Karthikeyan (2022)* detect tampered images by extracting many features from these images using Speeded Up Robust Features (SURF), Local Binary Pattern (LBP), enhanced LBP, and enhanced SURF. Particle Swarm Optimization (PSO) is introduced to combine all these features to select only the most significant features. Finally, the classification stage is performed using a combination of three classification techniques: support vector machine (SVM), back propagation neural networks (BPNN), and Ensemble. *Alahmadi et al. (2017)* present a system to detect copy–move and splicing forgeries based on local binary pattern (LBP) and discrete cosine transform (DCT). Initially, the RGB images are transformed to YCbCr form to get the chroma channels that hold most of the tampering traces undetected by the naked eye. The image is divided into overlapping blocks, and then a local binary pattern (LBP) and a 2D discrete cosine transform are employed to model the tampering traces and extract discriminative features. In the end, a support vector machine is used for

classification. *Saeed, Hamid & Ahmed (2023)* propose a copy-move recognition technique based on dividing the image into blocks and extracting features from each block using the Gabor filter. Principal component analysis is utilized after that to reduce the features. Finally, a matching step is performed to detect duplicated blocks.

The handcrafted methods are computationally efficient but cannot extract the whole texture features, which yields a misclassification (*Nazir et al., 2022*). In recent years, deep learning methods attracted the attention of researchers for detecting copy-move and image splicing because they extract more details about the images, which improves the classification accuracy (*Nguyen et al., 2022*; *Chakraborty, Chatterjee & Dey, 2022*; *Qazi, Zia & Almorjan, 2022*; *Ding et al., 2023*; *Zhuang et al., 2021*; *Gupta et al., 2022*; *Ali et al., 2022*). *Qazi, Zia & Almorjan (2022)* introduce a tampering detection algorithm based on the state-of-the-art ResNet50v2 that identifies various types of image manipulations. The algorithm fine-tunes the ResNet50v2 architecture using the weights of a YOLO convolutional neural network (CNN) to identify manipulated images accurately. *Ganguly et al. (2022)* propose a soft attention-based technique that distinguishes original images from tampered ones where the person's face is replaced with a different person's face. The process starts by extracting the face region using a Multi-Task Cascading Neural Network (MTCNN) model, and then a sliced Xception model is applied to the extracted faces to generate the feature maps. Subsequently, a soft attention mechanism declares the relevant features that discriminate against the properties of a fake face. In the end, a classification step is added to classify the images as real or fake. *Walia et al. (2022)* present a system for detecting manipulated images based on analyzing the difference in JPEG compression levels and integrating it with the texture information of the image using a LBP. Shapley additive explanation (SHAP), which is an explainable artificial intelligence (XAI) approach, is used as a feature selection strategy for the generated feature map from scale-invariant and direction-invariant LBP (SD-LBP). Finally, a ResNet50 model is fine-tuned on the forgery datasets using the selected features to classify original and forged images. *Hammad, Ahmed & Jamil (2022)* introduce a copy-move forgery detection system that utilizes the AlexNet deep learning model for feature extraction from suspicious images. Following feature extraction, a feature selection algorithm called ReliefF is applied to identify the most significant features. Finally, a logistic classifier is employed to identify forged images.

*Ali et al. (2022)* design a CNN-based model to differentiate between genuine and forged images. The system depends on the fact that the original and the forged regions are from different sources, so their compression rates differ. The algorithm exploits this point by recompressing and subtracting the recompressed suspected image from the original suspected image. The resultant difference trains the CNN model to identify authentic and forged images. *Chaitra & Reddy (2023)* propose a copy-move detection approach that first uses YOLO to detect all objects in an image. Afterward, VGG Net is applied to each object to extract representative features. These features are then fed to a Deep CNN to identify multiple forgeries in this image. Besides, a fully convolution neural network called MiniNet is proposed by *Tyagi & Yadav (2023b)* to detect splicing and copy-move. The network consists of four convolutional layers, four max pool layers of size ($2 \times 2$), two fully

connected layers, a dropout layer, and a classification layer. Moreover, *Tyagi & Yadav (2023a)* present a new CNN called ForensicNet designed to identify various types of image manipulations, such as copy-move, splicing, and retouching. ForensicNet Architecture uses the inverted bottleneck technique inspired by transformers to improve accuracy and reduce network parameters. It also utilizes separate downsampling layers inspired by ResNets for faster network convergence. Mixing information in the spatial dimension is performed using depth-wise convolutions that depend on MobileNetV2. *Mallick et al. (2022)* propose a copy-move and splicing detection algorithm that applies ELA to the images as a preprocessing step and learns VGG16 and VGG19 models using the ELA output. *Muniappan et al. (2023)* build a CNN model consisting of two convolutional layers, two max-pooling layers, a dense layer, and an output layer to detect image manipulation types. The algorithm utilizes error-level analysis to train the CNN model to differentiate between original and fake images. *Walia et al. (2021)* combine two streams of features for detecting image forgery. Stream one extracts the handcrafted features that represent color characteristics by segmenting the RGB images into blocks, and then the feature vector is created utilizing intra-block and inter-block information. The second stream extracts the luminance channel from the YCbCr of the image, which is then used to build a Local Binary Pattern (LBP) of the image. Afterward, LBP is fed to a ResNet-18 network to construct a feature vector. The resultant fused vector of the two streams is normalized and provided to a shallow neural network (SNN) for forgery classification.

## METHODOLOGY

This research proposes a CNN-based approach for detecting image manipulations. The proposed method exploits the pros of ELA, which reveal discrepancies that indicate the modified regions of the image, and integrates it with CNN to effectively extract the feature that reflects the modified regions. During the training phase, the ELA algorithm is employed on the training images to identify areas that have been tampered with. Subsequently, the developed CNN model undergoes training on these modified images to discern between genuine and altered images. In the testing phase, the ELA technique is applied to the test images, and the resultant data is inputted into the trained model for classification. Figure 1 introduces the proposed method's basic architecture, and the following subsections describe each part in detail. Furthermore, Algorithm 1 provides a brief overview of the procedural steps comprising the ELA-CNN manipulation detection model to facilitate a clear understanding of the sequential operations executed by the algorithm.

### Error level analysis

Error level analysis (ELA) is a well-known technique for determining areas in an image exposed to tampering, especially image splicing and copy-move forgeries. The ELA idea depends on exploring regions with different compression levels; this variation in compression levels may result from subjecting parts of an image to various types of lossy compressions or repeatedly using the same compression type for various areas of the image. The process begins by subjecting the suspected image to additional lossy

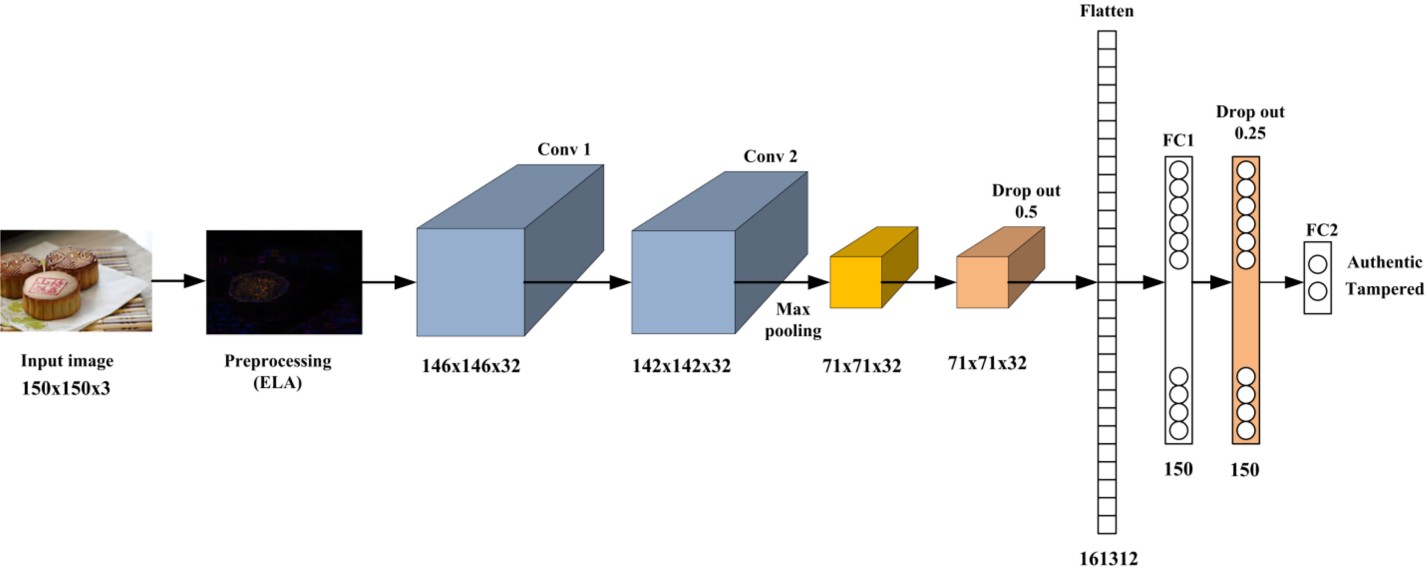

**Figure 1** The proposed image forgery framework.

---

**Algorithm 1  The proposed ELA-CNN image manipulation detection algorithm**

**Input** dataset, epochs, batch_size, new_test_image

**Output** trained_model, evaluation criterion, predicted_image_class (*i.e.*, authentic or manipulated)

1: ela_images = imagetoELA(dataset)                              ▷Dataset preparation step

2: [X, Y] = train_test_split(ela_images)

3: *model* = Sequential()                                        ▷CNN model architecture step

4: *model*.add(Conv2D(32, (5, 5), 'valid', 'relu', (150, 150, 3)))

5: *model*.add(Conv2D(32, (5, 5), 'valid', 'relu'))

6: *model*.add(MaxPool2D((2, 2)))

7: *model*.add(Dropout(0.25))

8: *model*.add(Flatten())

9: *model*.add(Dense(150, 'relu'))

10: *model*.add(Dropout(0.5))

11: *model*.add(Dense(2, 'sigmoid'))

12: *model*.compile(optimizer='adam', loss='binary cross entropy')  ▷CNN model training step

13: *model*.fit(x_train, y_train, epochs = epochs, batchsize = batch_size, validation=(x_val, y_val))

14: loss, accuracy, precision, recall = *model*.evaluate(x_test, y_test)

15: ela_new_image = imagetoELA(new_test_image)                    ▷CNN model testing step

16: predicted_image_class = *model*.predict(ela_new_image)

---

compression and subtracting the output from the original suspected image to highlight the tampered areas, resulting in a higher ELA value. The pros of the ELA are that it amplifies error levels, making discrepancies simpler to identify and analyze. This visualization assists

forensic specialists in identifying prospective areas of investigational interest. Therefore, ELA is integrated with the CNN model to automatically classify manipulated images as authentic or tampered images. The basic steps involved in ELA are as follows *Sudiatmika et al. (2019)*, *Kumar, Chowdhary & Srivastava (2021*, *2020)*:

1. Compress the original image using a lossy compression technique, such as JPEG, with low compression settings. This technique introduces new compression artifacts while maintaining current ones.

2. Calculate the error levels by subtracting the compressed image from the original image to produce a difference image. This difference image represents the error levels presented during the compression.

3. Apply algorithms such as rescaling or high-pass filtering to intensify the error levels in the difference image. This intensification makes the inconsistencies more visible.

4. Observe the intensified error levels in the difference image. Areas with higher error levels reflect possible manipulations or alterations.

## The proposed CNN model

Over the last few years, CNNs have exhibited outstanding performance on tasks such as image classification. CNNs are designed to autonomously learn and extract related features from input images to address a wide range of computer vision tasks effectively. CNNs utilize convolutional and pooling layers to efficiently detect hierarchical patterns and spatial correlations within the input image to represent complex visual details (*Zafar et al., 2022*; *Patil & Rane, 2021*). This research exploits the advantages of CNN and constructs a CNN model that consists of two convolutional, one max pooling, two dropouts, and two fully connected layers, as shown in Fig. 2. First, an input image with dimensions $150 \times 150 \times 3$ is fed into two consecutive convolution layers with 32 filters of kernel size $5 \times 5$ and the rectified linear unit (ReLU) as an activation function to generate a hierarchical feature map of $142 \times 142 \times 32$. The two consecutive convolutional layers enable the model to learn hierarchical representations of the input image. Each convolutional layer captures different levels of abstraction, starting from low-level features like edges and textures to higher-level features such as shapes and structures associated with image manipulations. The max pooling layer with a filter size $2 \times 2$ following the two consecutive convolutional layers helps summarize and downsample these learned features to create a more compact and informative representation of the feature map with a size of $71 \times 71 \times 32$. Max pooling is selected because it can be beneficial for detecting localized manipulations in an image. Max pooling emphasizes the input image's most salient features or regions by choosing the maximum value within each pooling region. If an image manipulation introduces a distinctive pattern or artifact in a localized region, max pooling captures and amplifies those features, making them more detectable.

Once the feature map dimension is down-sampled, it is fed to a dropout layer with a probability of 0.25 to learn more robust and generalizable features. This reduces the risk of

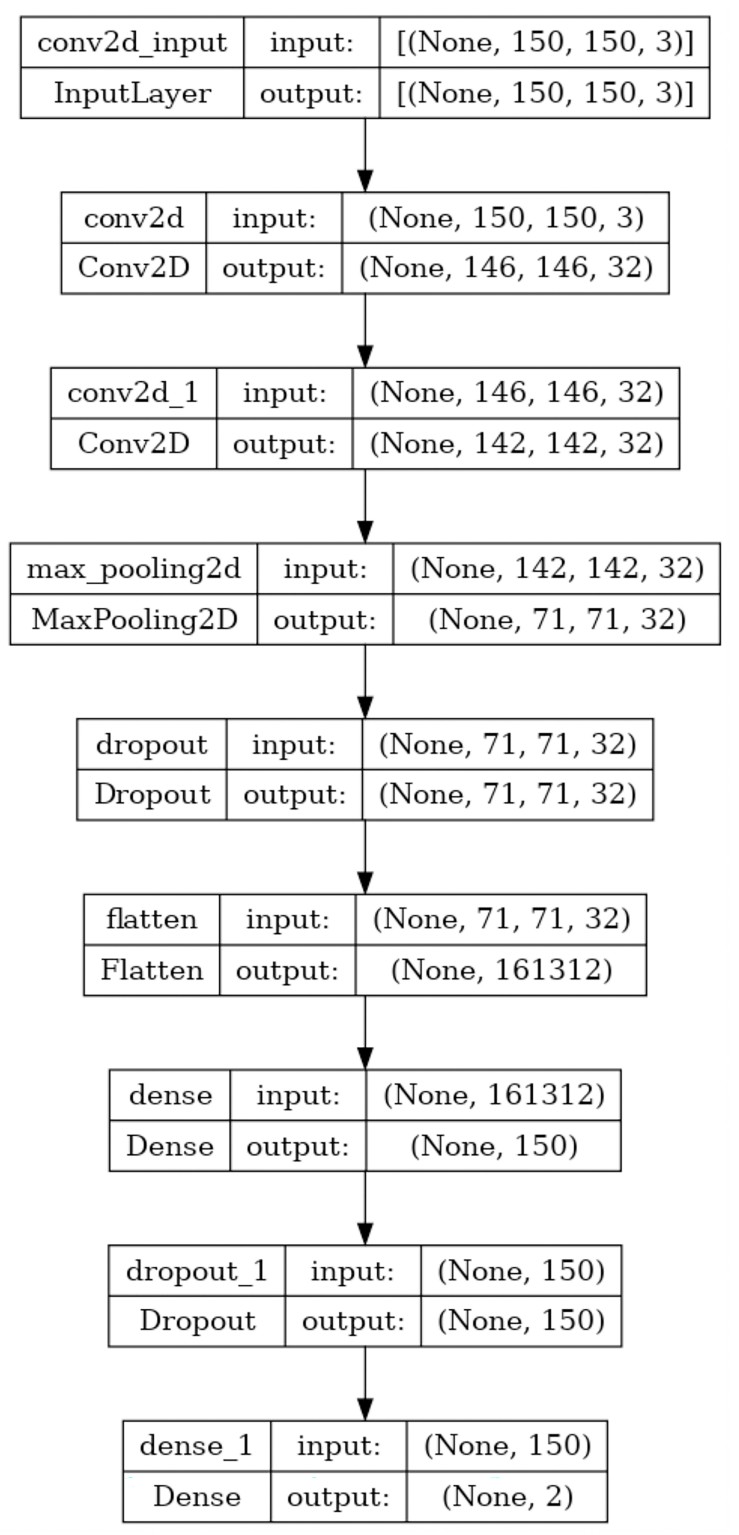

**Figure 2 The proposed CNN model architecture.**

**Table 1 The model architecture includes the filter size, feature map size, and activation function.**

| Layer | Filter size | Feature map size | Activation function |
|---|---|---|---|
| Input | – | $150 \times 150 \times 3$ | – |
| Conv1 | $5 \times 5$ | 146, 146, 32 | ReLU |
| Conv 2 | $5 \times 5$ | 142, 142, 32 | ReLU |
| Max pooling 2 | $2 \times 2$ | 71, 71, 32 | |
| Drop out | – | 71, 71, 32 | |
| Flatten | – | 161,342 | – |
| Fully connected 1 | – | 150 | ReLU |
| Drop out 1 | – | 150 | – |
| Fully connected 2 | – | 2 | Sigmoid |

overfitting the training data and improves the model's ability to generalize to unseen images.

The next layer in the proposed image manipulation model is a dense layer of 150 neurons. This dense layer uses the ReLU activation function, which introduces non-linearity to the network and helps capture complex patterns and features. A dropout layer with a probability of 0.25 is applied to the dense layer output to drop out a fraction of the neuron activations during training. The output of this dropout layer is then connected to two neurons, representing the output neurons of the model. These output neurons use the sigmoid activation function, which squashes the output values between 0 and 1.

The sigmoid activation is commonly used in binary classification tasks, where each neuron represents a class (authentic or forged) and provides a probability estimate of the input image belonging to that class. Table 1 summarizes the model architecture regarding the filter size, feature map size, and the activation function used in each layer.

# EXPERIMENTAL RESULTS

This section presents the dataset and details of the training hyperparameters. Moreover, the efficiency of the proposed method is evaluated based on the selected dataset and compared with state-of-the-art algorithms.

## Dataset and training hyperparameters

The proposed method is trained and evaluated on the CASIA 2.0 dataset (*Dong, Wang & Tan, 2013*). This dataset comprises 12,614 images in JPEG and TIFF formats with sizes 320 × 240 and 800 × 600, 7,491 authentic, and 5,123 manipulated images. Animals, architecture, art, interiors, nature, plants, and texture are just a few of the themes represented in these images. Copy move and image splicing are the manipulation methods in CASIA 2, 3,274 of which are copy-move and 1,849 are image splicing, and these images are created using Adobe Photoshop. The proposed method is only applied to the JPEG images, whose numbers are 9,501. The dataset is split into 80% training and 20% validation and testing. The experiments are conducted on the Kaggle platform using Keras with a TensorFlow backend. The specific configuration provided by Kaggle includes four CPU

cores and 30 GB of RAM. We used a batch size of eight and trained the models for 40 epochs, with a learning rate 0.0001 and a dropout probability of 0.25. We employed the Adam Optimizer and binary cross entropy as the loss function. These parameters are chosen based on preliminary experiments to optimize model performance.

## Evaluation metrics

The efficiency of the proposed model is evaluated based on three metrics: accuracy, precision, and recall, which are calculated mathematically from Eqs. (1)–(3), respectively.

$$Accuarcy = \frac{TN + TP}{TP + FP + TN + FN} \tag{1}$$

$$Precision = \frac{TP}{TP + FP} \tag{2}$$

$$Recall = \frac{TP}{TP + FN} \tag{3}$$

where TP is the true positive number, TN is the true negative number, FP is blackgray zero the false positive number, and FN is the false negative number. Therefore, accuracy provides an overall evaluation of the classifier's effectiveness across all classes. Precision measures how accurately the classifier recognizes positive cases. A higher precision rating indicates fewer false positives. Recall measures a classifier's ability to identify all positive instances. It shows how well the classifier identifies positive examples. A better recall value indicates fewer false negatives.

## Simulation results

From the perspective of assessing the performance of the proposed model, the confusion matrix is used to provide a comprehensive and informative summary of the classification performance of this model as applied in *Bukhari et al. (2021,2022)*. Table 2 presents the confusion matrix of the proposed algorithm. It can be noticed from the table that 90% of the forged images are predicted to be forged, while the model misclassifies 10% of them as authentic images. Moreover, 95.5% of the authentic images are classified as authentic, while 4.5% are predicted to be forged.

For assessing the model visually, Fig. 3 demonstrates examples of successfully detecting authentic images and their corresponding ELA. The figure shows that the ELA images of the authentic image are approximately black, which means that the image has not been subjected to any manipulations, and the proposed model correctly classifies those images.

Moreover, examples of successfully detecting forged images are presented in Fig. 4A, and their corresponding ELA and ground truth are introduced in Figs. 4B and 4C, respectively. It can be observed from the ELA images that the tampered images succeed in identifying the areas within the forged image with different compression levels, reflecting that ELA discriminates between the forged and authentic images correctly and helps the CNN model to be trained to differentiate between them.

On the other hand, according to the confusion matrix, the proposed model misclassifies 10% of the forged images and considers them authentic. Figure 5 shows samples of these misclassified images. It can be perceived from the figure that the ELA of the tampered

**Table 2 Confusion matrix for assessing the classification accuracy of the proposed model.**

| True | Forged | 90% | 10% |
|---|---|---|---|
| | Authentic | 4.5% | 95.5% |
| | | Forged | Authentic |
| | | Predicted | |

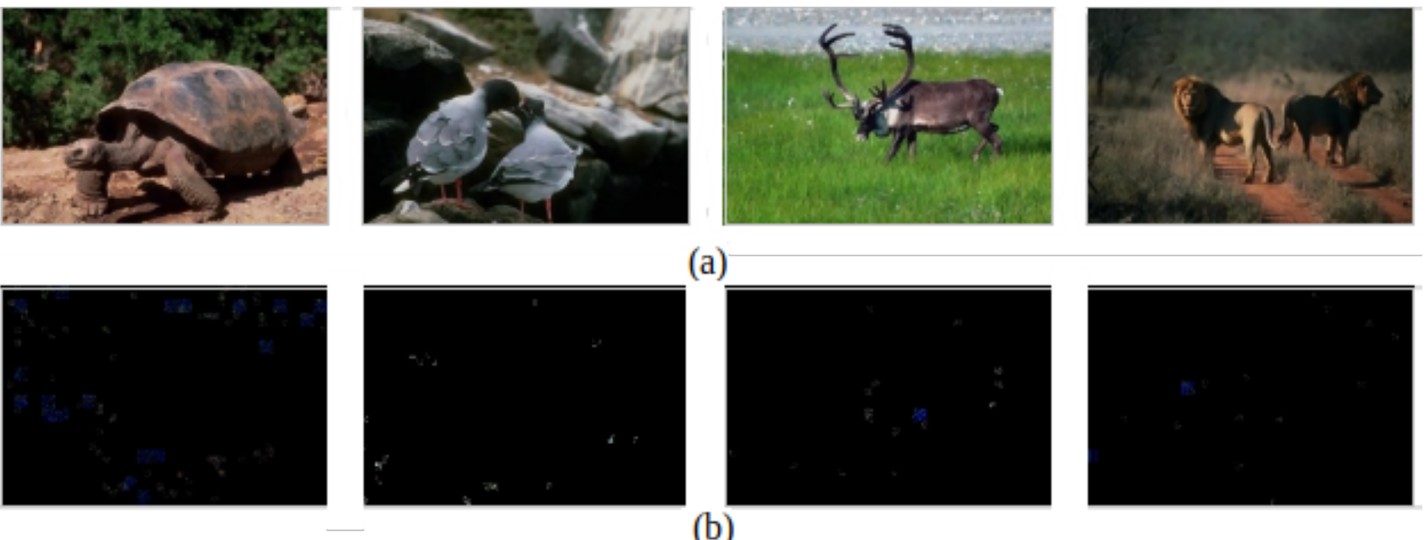

**Figure 3 Successfully detecting authentic images.** (A) Original images. (B) The corresponding ELA images.

regions is not clear visually, which may confuse the CNN model when classifying these images correctly.

Besides, the proposed algorithm is compared to the state-of-the-art algorithms mentioned in Table 3 regarding training accuracy, testing accuracy, precision, and recall. It can be observed from the results that the proposed model performs better than *Muniappan et al. (2023)*, *Tyagi & Yadav (2023a)*, and *Ali et al. (2022)* by 2 to 10 in terms of training accuracy, while its training accuracy is approximately equal to that of *Walia et al. (2022)*, *Walia et al. (2021)*, and *Qazi, Zia & Almorjan (2022)*. Further, the proposed algorithm outperforms all the conventional methods in testing accuracy, precision, and recall. Certain results may be missed because the findings are from the algorithms' original publications. Remarkably, our investigation has uncovered that, despite the less complex structure of our model, the CNN model proposed in this study has attained accuracy levels that are comparable to those achieved by the ResNet 50 model employed in the research conducted by *Qazi, Zia & Almorjan (2022)* and *Walia et al. (2022)*. Furthermore, it is worth noting that the CNN model that has been proposed exhibits superior performance compared to the CAT-Net model introduced by *Kwon et al. (2021)*, particularly in terms of

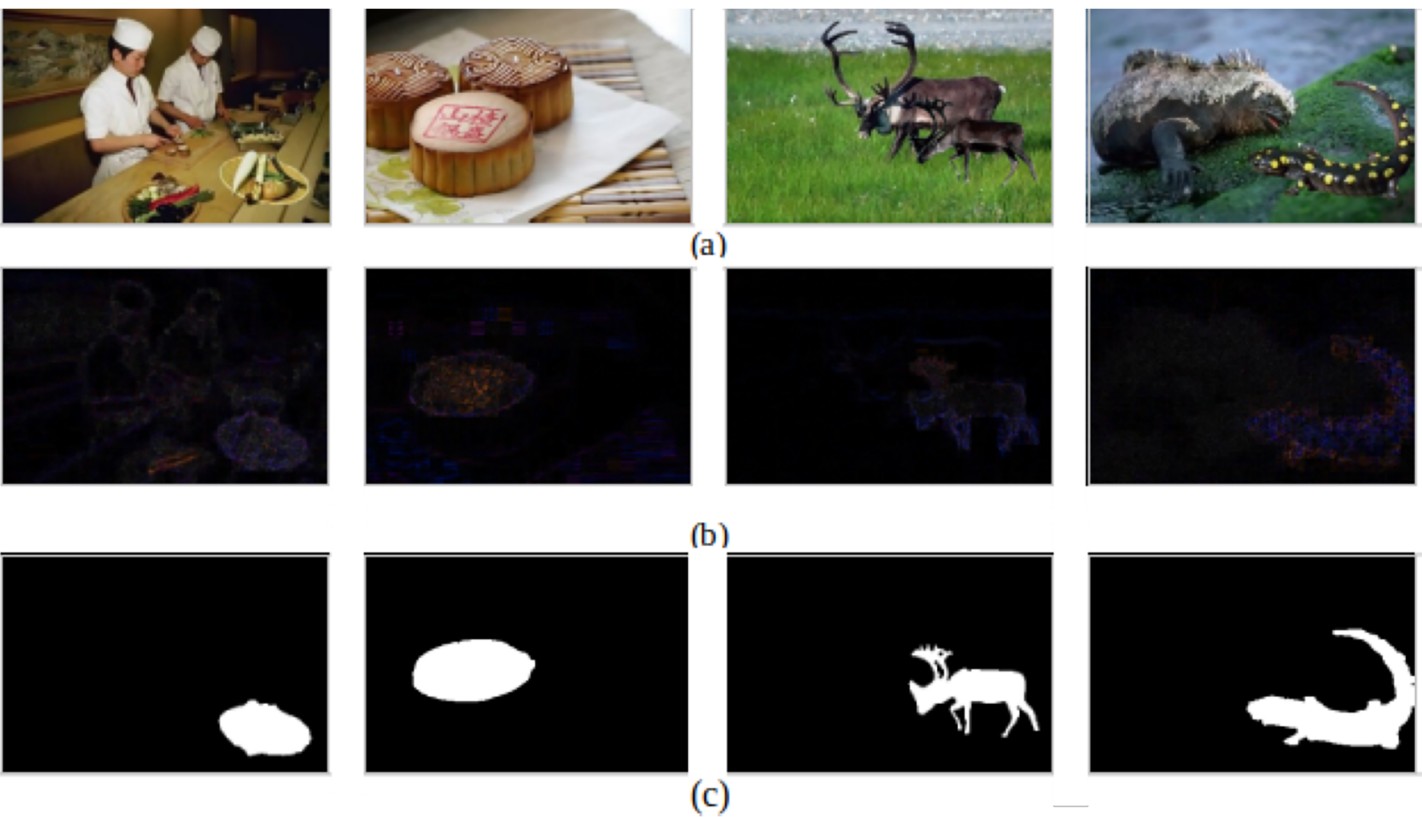

**Figure 4 Successfully detecting forged images.** (A) Forged images. (B) The corresponding ELA. (C) The corresponding ground truth.

precision, recall, and accuracy. Additionally, the proposed system's performance is evaluated based on inference time and memory utilization. Our findings indicate that the inference time for all test images is 52 s, with the model size approximately 277 megabytes. The findings above highlight the notable efficacy and proficiency of the proposed CNN methodology in the realm of image manipulation detection. It enables rapid identification of manipulated content to mitigate the spread of misinformation and protect the integrity of digital media in social media, journalism, and art.

## DISCUSSION

The proposed algorithm underwent several experiments to optimize its architecture. Thus, different configurations are adjusted and evaluated iteratively to identify the architecture that yields the best results for manipulation detection. The best five model configurations are selected and presented in Table 4. The architecture of each model with the layer arrangement is introduced in column 2. One of these configurations involved repeating the convolutional and max pooling layers four times, as in model 1. This adjustment resulted in a test accuracy of approximately 80%. Another experiment involved adding a batch normalization layer in conjunction with the convolutional and max pooling layers and repeating this group of layers as in models 2, 3, and 4. This modification led to an

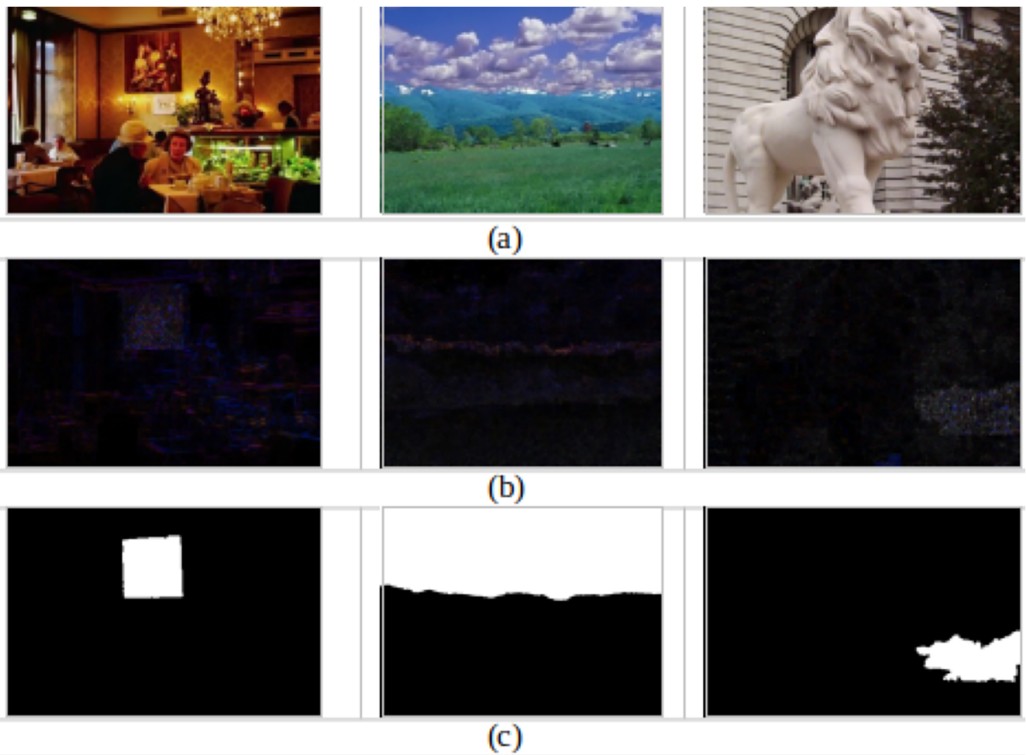

**Figure 5 Misclassified forged images.** (A) Forged images. (B) The extracted ELA. (C) The corresponding ground truth.

**Table 3 Comparison between the proposed model and the state-of-art techniques.**

| Method | Training accuracy | Testing accuracy | Precision | Recall |
|---|---|---|---|---|
| *Muniappan et al. (2023)* | 89 | – | 82 | 89 |
| *Tyagi & Yadav (2023a)* | 98.93 | – | 90.6 | 94.8 |
| *Kwon et al. (2021)* | – | 87.29 | 62 | 87 |
| *Ali et al. (2022)* | 94.93 | 92.23 | 85 | 97 |
| *Walia et al. (2022)* | 99.5 | – | – | – |
| *Walia et al. (2021)* | 99.3 | – | – | – |
| *Qazi, Zia & Almorjan (2022)* | 99.3 | – | – | – |
| The proposed technique | 99.05 | 94.14 | 94.1 | 94.07 |

**Table 4 Comparison between different CNN model architectures for the proposed model.**

| Model name | Architecture | Training accuracy | Test accuracy |
|---|---|---|---|
| Model 1 | 4 Conv-max pool flatten, Dense, Dropout dense | 98.55 | 80.37 |
| Model 2 | 4 Conv-batch norm-max pool dropout, Flatten, Dense dropout, Dense | 96.66 | 88.9 |
| Model 3 | 4 Conv-max pool-batch norm dropout, Flatten, Dense dropout, Dense | 90.18 | 86.8 |
| Model 4 | Conv, Max pool, Batch norm 3 Conv-max pool batch norm, Flatten, Dense dropout, Dense | 96.76 | 85 |
| The proposed model | 2 Conv-max pool, Dropout, Dense, Dropout, Dense | 99.05 | 94.14 |

improved test accuracy of 85 to 88%. The last configuration is the best and is selected as the proposed model because it achieves the best test and training accuracy with values of 94.14 and 99.05, respectively.

## CONCLUSION

This study presents a novel image manipulation detection system targeting copy-move and splicing forgeries. The proposed method leverages ELA to extract features and utilizes a new CNN architecture for robust classification. The network is constructed from two convolution layers, a max pooling layer, two dropout layers, and two fully connected layers. Evaluated on the CASIA 2.0 benchmark dataset, the system achieves high accuracy, precision, and recall, surpassing the performance of current state-of-the-art models. The impressive performance exhibited on the CASIA 2.0 dataset indicates promising potential for implementation in diverse fields where the integrity and authenticity of visual content are of the highest priority. While the system shows promising results in detecting image tampering, there are potential areas for future research. One avenue for investigation could focus on identifying the specific regions within an image that have been forged, providing more granular information about the tampering. Additionally, further research could explore methods to specify the type of tampering that has occurred, such as determining whether it is a copy-move or splicing forgery.

### Funding

This study is supported *via* funding from Prince Sattam bin Abdulaziz University project number (PSAU/2024/R/1445). The funders had no role in study design, data collection and analysis, decision to publish, or preparation of the manuscript.

### Grant Disclosures

The following grant information was disclosed by the authors:
Prince Sattam bin Abdulaziz University: PSAU/2024/R/1445.

### Competing Interests

The authors declare that they have no competing interests.

### Author Contributions

- Ahmad M. Nagm conceived and designed the experiments, performed the experiments, performed the computation work, prepared figures and/or tables, and approved the final draft.
- Mona M. Moussa analyzed the data, prepared figures and/or tables, authored or reviewed drafts of the article, and approved the final draft.
- Rasha Shoitan analyzed the data, prepared figures and/or tables, authored or reviewed drafts of the article, and approved the final draft.
- Ahmed Ali analyzed the data, authored or reviewed drafts of the article, and approved the final draft.

- Mohamed Mashhour conceived and designed the experiments, performed the experiments, performed the computation work, authored or reviewed drafts of the article, and approved the final draft.
- Ahmed S. Salama conceived and designed the experiments, analyzed the data, prepared figures and/or tables, and approved the final draft.
- Hamada I. AbdulWakel conceived and designed the experiments, performed the experiments, performed the computation work, authored or reviewed drafts of the article, and approved the final draft.

### Data Availability

The CASIA 2.0 Image Tampering Detection Dataset analyzed during the current study are available at Kaggle: https://www.kaggle.com/datasets/divg07/casia-20-image-tampering-detection-dataset.

The code is available at GitHub and Zenodo:

- https://github.com/researchschoolISA/CASIA2code.

- Mohamed_Mashhour. (2024). image-tampering-detection: Detecting Image Manipulation with ELA-CNN Integration (Detecting_Image_ELA-CNN). Zenodo. https://doi.org/10.5281/zenodo.11251149.

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
