# Peer review of "Detecting image manipulation with ELA-CNN integration: a powerful framework for authenticity verification"

_PeerJ Computer Science, doi:10.7717/peerj-cs.2205_

## Round 0.1 · original submission · Major Revisions

· Academic Editor

Major Revisions

After carefully considering the reviews and assessing your manuscript, I am pleased to invite you to revise and resubmit your manuscript for further consideration. The reviewers have provided constructive comments that will help strengthen your work. Please address each of these points thoroughly in your revised manuscript. Additionally, ensure that you provide a detailed response letter outlining how you have addressed each comment raised by the reviewers. This will help the reviewers and myself evaluate the changes made to the manuscript. Note that it is PeerJ's policy that additional references suggested during the peer-review process should only be included if the authors agree that they are relevant and useful.

Reviewer 1 ·

Basic reporting

'no comment

Experimental design

Discuss the robustness of your proposed method against common image manipulations beyond copy-move and splicing, such as resizing, rotation, or blurring.
What is the computational efficiency of your algorithm, particularly in terms of inference time and memory requirements
Provide more details about the specific hyperparameters of CNN model, such as the number of filters, kernel size, and dropout probability.
Include other up to 6 confusion matrix-based evaluation metrics ..cite the following here

PMID: 34829338,PMID: 35215090 Provide more context on the significance of copy-move and splicing forgeries in real-world scenarios
Discuss the implications of false positives and false negatives based on the confusion matrix in the context of image manipulation detection
Discuss any benchmark technique and compare your proposed model with that
Infact pre-trained networks should have been used and compared with
What is the overall applicability of this study. Mention this as a separate section.
Add the literature review section as a separate section from the introduction.
Introduction section is very small..increase it.

Validity of the findings

Authors need to add a separate Comparative analysis section and compare their proposed model with the benchmark studies or other techniques

Reviewer 2 ·

Basic reporting

The study aimed to use artificial intelligence algorithms to detect forgery in images and was developed around this idea. Although the idea is generally good, there are serious deficiencies in the paper.
1) Abstract is insufficient in terms of containing the results. What are the results and percent error rates?
2) Information about the general structure of the study is complex and insufficient.
3) Why was only CNN chosen? CNN is an old algorithm in image processing.
4) How are the parameters used for algorithms determined? Has a Preliminary Study been conducted?
5) The effects of results of the study need to be analyzed in detail.
6)Conclusion section is very inadequate. The results of such a study need to be better conveyed to the reader.
7) On what basis was the performance evaluation made? What metrics were used? Why is only the accuracy rate shown? There is no detailed information about these.

Experimental design

1) Why was only CNN chosen? There are lots of machine and deep learning algorithms. CNN is an old algorithm in image processing. Did authors used another algorithms?
2) How are the parameters used for algorithms determined? Has a Preliminary Study been conducted? Or they took from another paper? They need to be more clear about that.
3) The effects of results of the study need to be analyzed in detail. They should add discussion section.
4)Conclusion section is very inadequate. The results of such a study need to be better conveyed to the reader.
5) On what basis was the performance evaluation made? What metrics were used? Why is only the accuracy rate shown? There is no detailed information about these.

Validity of the findings

As above

---

## Round 0.2 · Minor Revisions

· Academic Editor

Minor Revisions

One of the reviewers suggested minor editing related to English language, and environment parameters. You are required to address these and resubmit.

Reviewer 1 ·

Basic reporting

All the comments have been addressed by authors and manuscript looks in good shape now. May accept it as is.

Experimental design

no comment

Validity of the findings

no comment

Additional comments

no comment

Reviewer 3 ·

Basic reporting

The authors improved this paper. However, there are mistakes in the English and grammar. It should be checked carefully.

Experimental design

It is improved. Please add the simulation environment, parameters, and the details of the simulation tool in the experiment section.

Validity of the findings

It is fine.

---

## Round 0.3 · accepted · Accept

· Academic Editor

Accept

I am pleased to inform you that your paper has been accepted for publication in PeerJ Computer Science. Your manuscript has undergone rigorous peer review, and I am delighted to say that it has been met with high praise from our reviewers and editorial team. Your research makes a significant contribution to the field, and we believe it will be of great interest to our readership. On behalf of the editorial board, I extend our warmest congratulations to you.